# High Efficiency and High Voltage Conversion Ratio Bidirectional Isolated DC–DC Converter for Energy Storage Systems

Yu-En Wu *[ID] and Kuan-Chi Chen

Department of Electronic Engineering, National Kaohsiung University of Science and Technology, Kaohsiung City 805301, Taiwan
* Correspondence: yew@nkust.edu.tw; Tel.: +886-7-6011000

**Abstract:** In this paper, a novel high-efficiency bidirectional isolated DC–DC converter that can be applied to an energy storage system for battery charging and discharging is proposed. By integrating a coupled inductor and switched-capacitor voltage doubler, the proposed converter can achieve isolation and bidirectional power flow. The proposed topology comprises five switches and a common core coupled inductor that uses only a set of complementary pulse-width-modulated signals to control and achieve high voltage gain without requiring high turn ratios or excessive duty cycles. Moreover, the proposed topology can recover the leakage inductance energy to improve the conversion efficiency. The main switches exhibit zero-voltage switching, which reduces the switching losses. A 500-W bidirectional converter is used to verify the feasibility of the proposed bidirectional converter through theoretical analysis and experiments. The experimental results indicate that the highest efficiency of the proposed converter in the step-up and step-down modes is 97.59% and 96.5%, respectively.

**Keywords:** bidirectional DC–DC converter; three-winding coupled inductor; zero-voltage switching (ZVS)

## 1. Introduction

Since the industrial revolution, fossil fuels have been used for generating power. The resulting emissions have caused serious damage to the environment. Therefore, countries worldwide have realized the importance of renewable energy and advocate for renewable energy generation systems [1]. However, renewable energy generation varies with changes in the weather and environment. To overcome this problem, an energy storage system is required. When excess renewable energy is produced, the excess electrical energy can be stored in an energy storage system. This energy can then be used when the electricity demand peaks. A distributed generation system [2,3] is required to support a renewable energy system, as shown in Figure 1.

In an energy storage system, a DC–DC converter is required to transfer energy between a battery and a DC bus. DC–DC converters are of two main types: isolated converters and nonisolated converters. Common nonisolated bidirectional converters are derived from the boost converter, buck-boost converter, single-ended primary-inductor converter, and other unidirectional converters that exhibit advantages such as low cost, high stability, and high practicability. However, nonisolated bidirectional converters are limited by their duty cycle and low voltage conversion ratio. Therefore, switched-capacitor [4–6], coupled-inductor [7], and cascaded [8–10] converters are used to improve the voltage gain.

Existing isolated bidirectional converters are mostly derived from forward-flyback converters [11,12] and bridge converters [13–15]. Isolated bidirectional converters exhibit advantages such as high stability and practicability; however, their circuit design is more complicated than that of nonisolated bidirectional converters. Rapid advancements have been made in bidirectional DC–DC converter technology. In recent years, studies have

focused on increasing the conversion efficiency and reducing the number of components to improve the stability of converters.

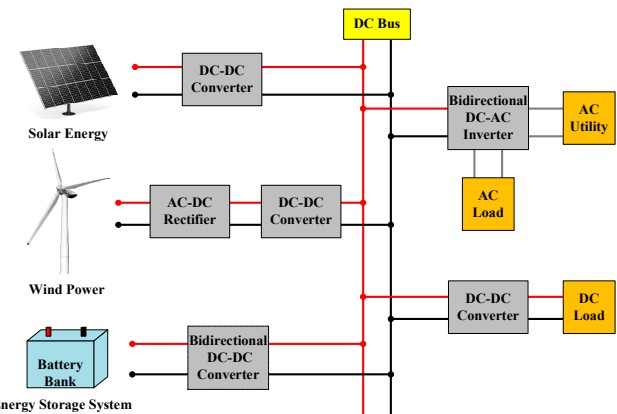

**Figure 1.** Configuration of a distributed generation system with an energy storage system.

　　A bidirectional buck-boost converter is a common topology that is mostly used in battery chargers [16–19]. A study proposed a nonisolated bidirectional quasi-Z-source DC–DC converter with high voltage gain; however, this converter does not have soft-switching technology and thus has a low conversion efficiency [20]. Another study developed a non-isolated interleaved converter with a three-phase interleaved switched-capacitor topology. This converter has advantages such as a high voltage gain and high power transmission; however, phase-shift control technology is required to control the aforementioned converter [21]. Coupled inductors have been incorporated into DC–DC nonisolated converters to increase the voltage gain of these converters [22–24]. Switches with zero-voltage switching (ZVS) can increase the conversion efficiency; however, a higher number of circuit components increases the cost of a converter [22,24]. Switches without ZVS require fewer components but exhibit higher losses than do those with ZVS [23]. A bidirectional forward-flyback converter with a simple circuit and low current ripple but a low voltage gain and complex control method was proposed in [25]. An isolated interleaved DC–DC converter with an interleaved topology that includes a voltage doubler (used to achieve high voltage gain and ZVS to increase efficiency) has been developed to reduce the current ripple; however, the circuit of this converter requires a complex control method and many components [26]. DC–DC converters with a wide input voltage range and high voltage gain have been developed [27,28]. One study used an additional active clamp to recycle the leakage inductor energy to increase the number of components and decrease the converter efficiency [27]. Another study used a current-fed topology to increase the output current of the low-voltage side of a converter to make this side suitable for battery charging; however, high current flows through the components of this converter, which results in it exhibiting high losses [28]. In contrast to the conventional dual-active-bridge converter, the bidirectional isolated bridge converter has been combined with a push–pull converter and full-bridge converter to reduce the number of switches required and thus the switching losses; however, the control method of the bidirectional isolated bridge converter is complex [29]. Another study used a three-winding coupled inductor and a half-wave voltage doubler to increase the voltage gain and lower the components of the circuit; however, the turns ratio of coupled inductor is high, which causes larger volume of the circuit [30]. In the present paper, a novel high-efficiency isolated DC–DC converter is proposed for an energy storage system. This converter can transfer energy between a battery and a DC bus. Since the common voltages of batteries and DC buses are 48 and 400 V, respectively, the low and high side voltages of the proposed converter are 48 and 400 V, respectively.

## 2. Circuit Architecture and Operation Principle

Figure 2 displays the circuit architecture of the proposed bidirectional converter. In this figure, $V_H$ and $V_L$ denote the high-voltage-side and low-voltage-side power ports, respectively. The coupled inductor consists of the leakage inductances $L_{lk1}$, $L_{lk2}$, and $L_{lk3}$ as well as the magnetizing inductance $L_{m1}$. Moreover, the turn ratio of the inductor is represented by $n$. The proposed converter contains five switches ($S_1$–$S_5$). The body diodes $D_{S1}$–$D_{S5}$ and parasitic capacitances $C_{S1}$–$C_{S5}$ are the parasitic elements of $S_1$–$S_5$, respectively. The proposed converter also contains the capacitors $C_1$–$C_4$. The operating principles and operation mode of the proposed topology in the step-up and step-down modes are analyzed.

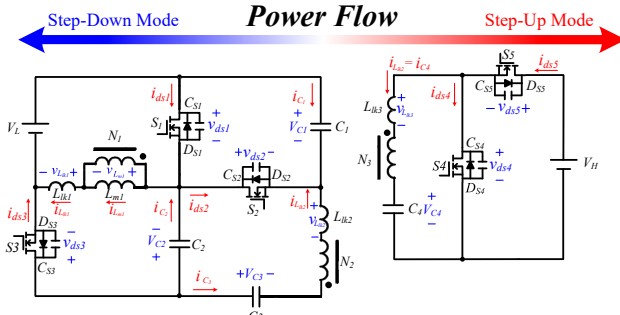

**Figure 2.** Voltage polarity and current direction of the components of the proposed topology.

This section discusses the operating principles of the proposed topology in the step-up and step-down modes. The voltage polarity and current direction of the components in the proposed topology are illustrated in Figure 2. To simplify our analysis of the operating principles, the following assumptions were made:

(1)   The internal resistance and parasitic effects can be ignored.
(2)   The voltages of the capacitors and currents of the inductors increase and decrease linearly.
(3)   The capacitances of $C_1$, $C_2$, $C_3$, and $C_4$ are infinite.
(4)   All the magnetic components operate in the continuous-current mode (CCM).
(5)   The number of turns $N_1 = N_2 < N_3$, and $N_2/N_1 = N_3/N_1 = n$.

### 2.1. Step-Up Mode

In the step-up mode, the complementary pulse-width-modulated (PWM) signal comprises two sets of signals: (i) $v_{gs1}$ and (ii) $v_{gs2,3}$. The gate signal of $S_1$ is the complementary waveform of $S_2$ and $S_3$. The signals of $S_4$ and $S_5$ are in the OFF state. The theoretical waveforms of the proposed topology in the step-up mode are displayed in Figure 3, and one operating cycle features five operation modes as shown in Figure 4a–e.

(1)   *Mode 1 [$t_0$–$t_1$]*

The equivalent circuit for the Mode 1 operation in the step-up mode is displayed in Figure 4a. This mode is operated in a dead-time period, and all switch signals are in the OFF state in Mode 1. At the beginning of this mode at time $t = t_0$, the low-voltage side $V_L$ and leakage inductance $L_{lk1}$ charge the capacitors $C_1$ and $C_2$. Mode 1 operation is followed by Mode 5 operation. To achieve ZVS, the energy of the parasitic capacitance of $S_1$ is released through $L_{lk2}$, and the energy of $L_{lk2}$ is recycled by $C_1$ and $C_2$. The energy of the magnetizing inductance $L_{m1}$ and $C_3$ is transferred to the high-voltage-side capacitor $C_4$ through the coupled inductor, and $L_{lk3}$ releases energy to $C_4$. Mode 1 ends when $S_1$ is turned on.

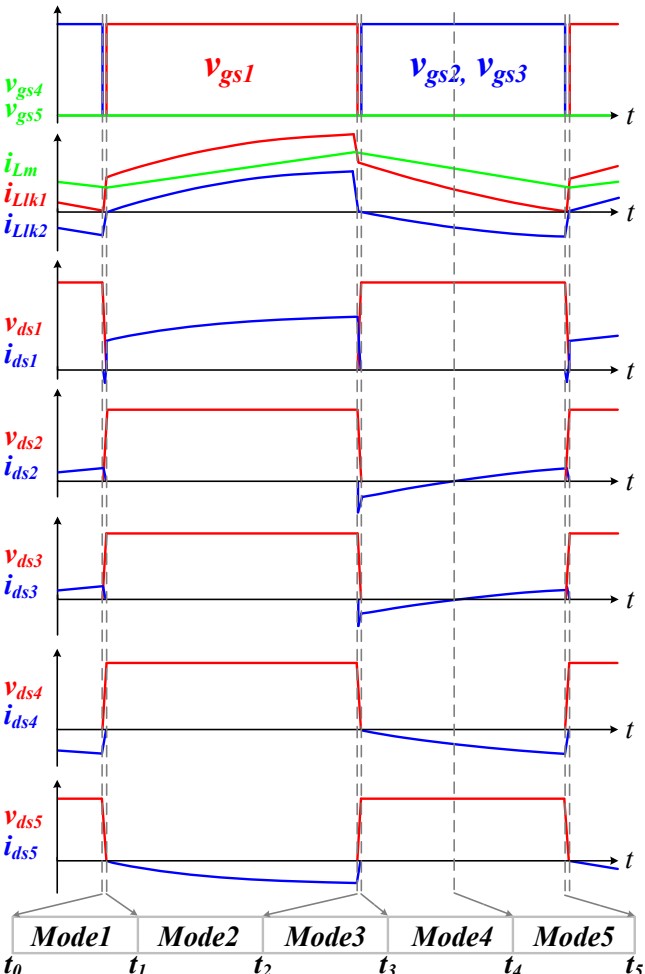

**Figure 3.** Key waveforms of the proposed topology in the step-up mode.

(2)    *Mode 2 [$t_1$–$t_2$]*

The equivalent circuit for Mode 2 operation is illustrated in Figure 4b. When $S_1$ is completely turned on at time $t = t_1$, the low-voltage side $V_L$ supplies energy to the magnetizing inductance $L_{m1}$ and leakage inductance $L_{lk1}$. Moreover, a part of the energy of $V_L$ is transferred to the high-voltage side $V_H$ through the coupled inductor. The capacitors $C_1$ and $C_2$ release energy to $C_3$ and the leakage inductance $L_{lk2}$, and a part of this energy is transferred to $V_H$ through the coupled inductor. The capacitor $C_4$ releases energy to $L_{lk3}$ and $V_H$.

(3)    *Mode 3 [$t_2$–$t_3$]*

The equivalent circuit for Mode 3 operation is depicted in Figure 4c. This mode is operated in a dead-time period, and all switch signals are in the OFF state in Mode 3. To achieve ZVS, the energy of the $S_3$ parasitic capacitance is absorbed into the leakage inductance $L_{lk1}$, and the energy of the $S_2$ parasitic capacitance is absorbed into the leakage inductance $L_{lk2}$. The energy of $L_{lk1}$ is released to $C_1$ and $C_2$, and the energy of $L_{lk2}$ is released to $C_2$ and $C_3$. The capacitor $C_4$ and leakage inductance $L_{lk3}$ release energy to $V_H$. Mode 3 ends when $S_2$ and $S_3$ are completely turned on.

(4)    *Mode 4 [$t_3$–$t_4$]*

The equivalent circuit for Mode 4 operation is displayed in Figure 4d. When $S_2$ and $S_3$ are completely turned on at time $t = t_3$, the leakage inductance $L_{lk1}$ continuously releases energy to $C_1$ and $C_2$. Capacitor $C_3$ charges $C_2$ and $L_{lk2}$. Moreover, a part of the energy of $C_3$ is transferred to $C_4$ and $L_{lk3}$ through the coupled inductor. The magnetizing inductance

$L_{m1}$ charges $C_4$ and $L_{lk3}$. Mode 4 ends when the capacitor voltage of $C_2$ is higher than the voltage of $N_1$.

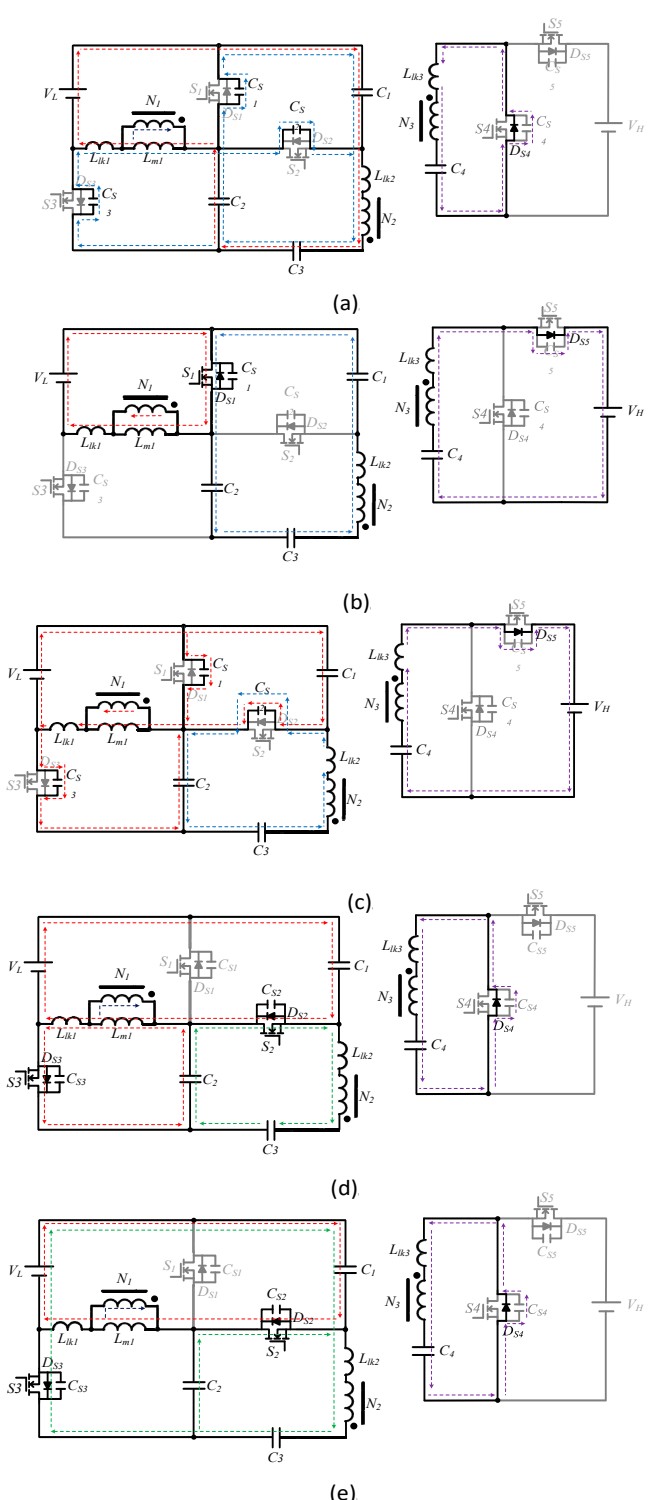

**Figure 4.** Equivalent circuits in the step-up mode: (**a**) Mode 1, (**b**) Mode 2, (**c**) Mode 3, (**d**) Mode 4, and (**e**) Mode 5.

(5)　*Mode 5 [$t_4$–$t_5$]*

The equivalent circuit for Mode 5 operation is illustrated in Figure 4e. At time $t = t_4$, the switch signals are the same as those in Mode 4. In Mode 5, $L_{lk1}$ releases en-

ergy to $C_1$, and $C_3$ charges $C_1$, $C_2$, and $L_{lk2}$. Moreover, a part of the aforementioned energy is transferred through the coupled inductor to $C_4$ and $L_{lk3}$. Mode 5 ends when the current of $L_{lk1}$ decreases to 0.

### 2.2. Step-Down Mode

In the step-down mode, the complementary PWM signal comprises two sets of signals: (1) $v_{gs1,5}$ and (2) $v_{gs2,3,4}$. The gate signals of $S_1$ and $S_5$ are complementary waveforms of those of $S_2$, $S_3$, and $S_4$. The theoretical waveforms of the proposed topology in the step-down mode are shown in Figure 5, and one operating cycle contains seven operation modes Figure 6a–g.

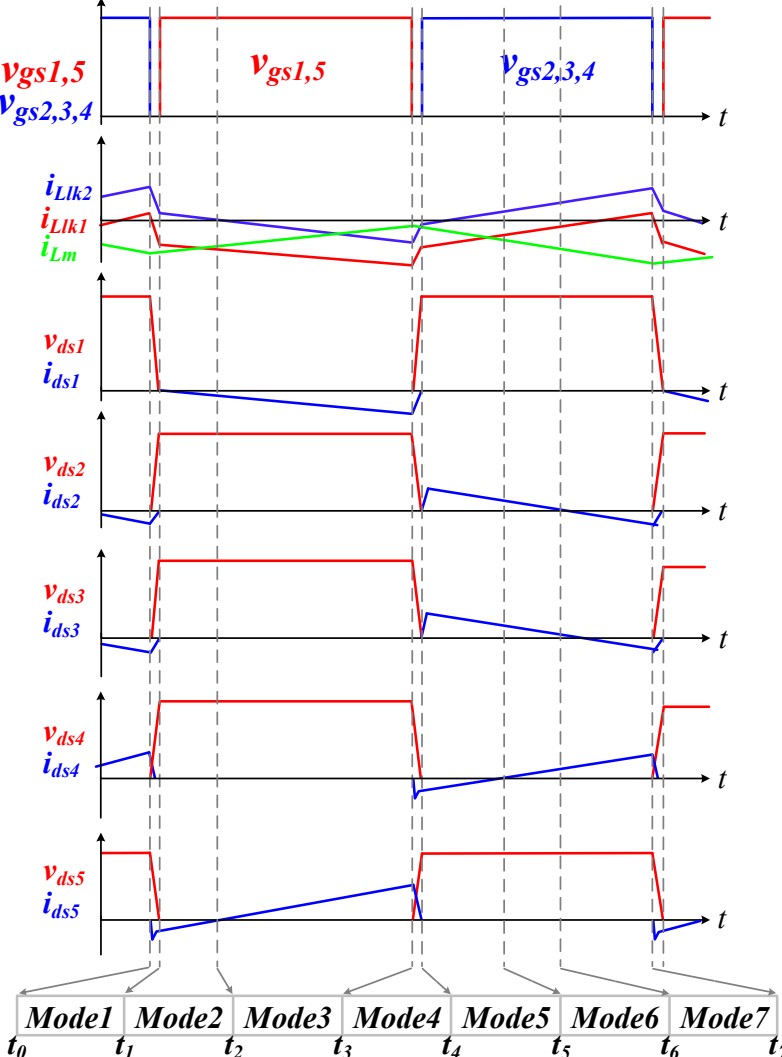

**Figure 5.** Key waveforms of the proposed topology in the step-down mode.

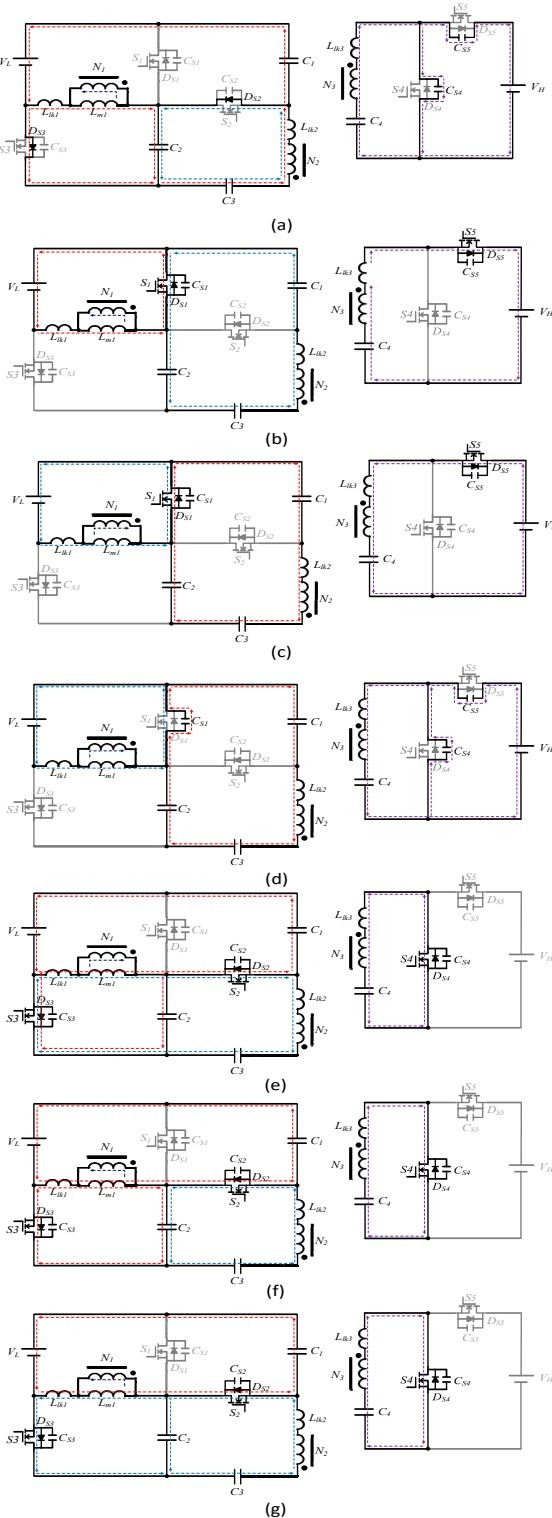

**Figure 6.** Equivalent circuits in the step-down mode: (**a**) Mode 1, (**b**) Mode 2, (**c**) Mode 3, (**d**) Mode 4, (**e**) Mode 5, (**f**) Mode 6, and (**g**) Mode 7.

(1)  *Mode 1 [$t_0$–$t_1$]*

The equivalent circuit for Mode 1 operation in the step-down mode is illustrated in Figure 6a. This mode is operated in a dead-time period, and all switch signals are in the OFF state. To achieve ZVS, the energy of the parasitic capacitance $S_5$ is absorbed into the leakage inductance $L_{lk3}$. Moreover, $C_1$ and the magnetizing inductance $L_{m1}$ charge the

low-voltage side $V_L$, $L_{lk1}$, and $C_3$. Mode 1 is followed by Mode 5. $C_2$ and $L_{lk2}$ release energy to $C_3$, and Mode 1 ends when $S_1$ and $S_5$ are completely turned on.

(2)   *Mode 2 [$t_1$–$t_2$]*

The equivalent circuit for Mode 2 operation is displayed in Figure 6b. When $S_1$ and $S_5$ are completely turned on at time $t = t_1$, $C_1$, $C_2$, and $L_{lk2}$ charge $C_3$ continuously. The magnetizing inductance $L_{m1}$ continuously transfers energy to $V_L$. The energy of $L_{lk3}$ and $C_4$ is released to the high-voltage side $V_H$. Mode 2 ends when the leakage inductance current $i_{Llk3}$ decreases to 0.

(3)   *Mode 3 [$t_2$–$t_3$]*

The equivalent circuit for Mode 3 operation is depicted in Figure 6c. At time $t = t_2$, the switch signals are the same as those in Mode 2. The high-voltage side $V_H$ provides energy to the low-voltage side $V_L$ through the coupled inductor, and the magnetizing inductance $L_{m1}$ charges $V_L$. Moreover, $C_3$ releases energy to $C_1$, $C_2$, and $L_{lk2}$.

(4)   *Mode 4 [$t_3$–$t_4$]*

The equivalent circuit for Mode 4 operation is shown in Figure 6d. This mode is operated in a dead-time period, and all switch signals are in the OFF state. To achieve ZVS, the energy of the parasitic capacitance $S_4$ is absorbed into the leakage inductance $L_{lk3}$. The leakage inductance $L_{lk1}$ charges $V_L$, and the energy of $L_{lk2}$ is released to $C_1$ and $C_2$. Moreover, $V_L$ continuously transfers energy to $V_L$ through the coupled inductor. Mode 4 ends when $S_3$ and $S_4$ are completely turned on.

(5)   *Mode 5 [$t_4$–$t_5$]*

The equivalent circuit for Mode 5 operation is displayed in Figure 6e. When $S_2$, $S_3$, and $S_4$ are completely turned on at time $t = t_4$, the capacitor $C_1$ and leakage inductance $L_{lk1}$ provide energy to $V_L$. The capacitors $C_2$ and $C_3$ charge $L_{lk1}$ and $L_{m1}$, and $L_{lk3}$ releases energy to $C_4$. Moreover, a part of this energy is transferred to $V_L$ through the coupled inductor. Mode 5 ends when the switch current of $S_4$ decreases to 0.

(1)   *Mode 6 [$t_5$–$t_6$]*

The equivalent circuit for Mode 6 operation is illustrated in Figure 6f. At time $t = t_5$, the switch signals are the same as those in Mode 5. The capacitor $C_1$ and leakage inductance $L_{lk1}$ continuously provide energy to $V_L$. The capacitor $C_2$ charges $L_{lk1}$, $L_{lk2}$, and $C_3$, and $C_4$ charges $L_{m1}$ through the coupled inductor. Mode 6 ends when the switch currents $i_{ds2}$ and $i_{ds3}$ decrease to 0.

(2)   *Mode 7 [$t_6$–$t_7$]*

The equivalent circuit for Mode 7 operation is shown in Figure 6g. At time $t = t_6$, the switch signals are the same as those in Mode 6. The capacitor $C_1$ and leakage inductance $L_{lk1}$ continuously provide energy to $V_L$. The capacitor $C_2$ charges $L_{lk2}$ and $C_3$, and $C_4$ continuously charges $L_{m1}$ through the coupled inductor. Mode 7 ends when $S_2$, $S_3$, and $S_4$ are completely turned off.

### 3. Steady-State Analysis

In the step-up mode, the complementary PWM signal comprises two sets of signals, the switching period is $D_1T_S$, $v_{gs1}$ is turned ON for time $D_1T_S$, and $v_{gs2,3}$ is turned ON for time $(1 - D_1)T_S$.

In the step-down mode, the complementary PWM signal comprises two sets of signals, $v_{gs1,5}$ is turned ON for time $D_3T_S$, and $v_{gs2,3,4}$ is turned OFF for time $(1 - D_3)T_S$. The steady-state analysis of the proposed topology is based on the following assumptions:

(1)   All internal resistances and parasitic effects are ignored.
(2)   The currents of the inductors and voltages of the capacitors increase and decrease linearly.
(3)   $N_2/N_1 = n$.

(4) All magnetic components are operated in the CCM.

(5) The capacitances of $C_1$, $C_2$, $C_3$, and $C_4$ are infinite.

### 3.1. Step-Up Mode

(1) *Voltage Gain Analysis*

On the basis of Kirchoff's voltage law (KVL), the voltage of the magnetizing inductance $V_{Lm1}$ for time $D_1 T_S$ can be expressed as follows:

$$V_{Lm1} = V_L = V_{c3} - V_{c2} - V_{c1} + V_L + \frac{V_H - V_{c4}}{n} = L_{m1} \frac{\Delta i_{Lm1,on}}{D_1 T_S} \tag{1}$$

For time $(1 - D_1)T_S$, the voltage of the magnetizing inductance $V_{Lm1}$ can be expressed as follows:

$$V_{Lm1} = V_L - V_{c1} = -V_{c2} = -\frac{V_{c4}}{n} - V_{c3} = L_{m1} \frac{\Delta i_{Lm1,off}}{(1 - D_1) T_S} \tag{2}$$

When volt-second balance is achieved for the inductor at time $(1 - D_1)T_S$, the following equation is satisfied:

$$\Delta i_{Lm1,on} = \Delta i_{Lm1,off} \tag{3}$$

By substituting (1) and (2) into (3), the following expressions are obtained for the voltages of $C_1$, $C_2$, $C_3$, and $C_4$:

$$V_{C1} = \frac{1}{(1 - D_1)} V_L \tag{4}$$

$$V_{C2} = \frac{D_1}{(1 - D_1)} V_L \tag{5}$$

$$V_{C3} = \frac{1}{(1 - D_1)} V_L \tag{6}$$

$$V_{C4} = \frac{ND_1}{(1 - D_1)} V_L \tag{7}$$

In Mode 2 of the step-up mode, the relationship among $V_H$, $V_{C4}$, and $V_L$ is as follows:

$$V_H = V_{c4} + nV_L \tag{8}$$

By substituting (7) into (8), the voltage gain in the step-up mode ($G_{step-up}$) is obtained as follows:

$$G_{step-up} = \frac{V_H}{V_L} = \frac{n}{(1 - D_1)} \tag{9}$$

Figure 7 presents the relationship between the voltage gain and the duty cycle in the step-up mode.

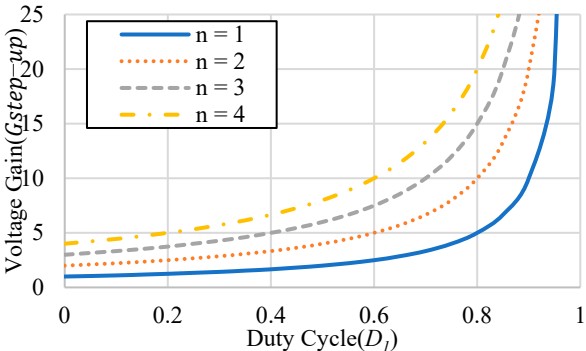

**Figure 7.** Relationship between the voltage gain and the duty cycle in the step-up mode.

(2)    *Voltage Stress Analysis*

According to the equivalent circuit at time $D_1T_S$, the voltage across $S_2$ is $V_{C1}$, and the voltage across $S_3$ is the sum of $V_{C2}$ and $V_L$. The voltage stress of $S_4$ is $V_H$. The voltage stresses of the switches at the aforementioned time are expressed as follows:

$$V_{S2,stress} = V_{C1} = \frac{1}{1-D_1}V_L = \frac{1}{n}V_H \tag{10}$$

$$V_{S3,stress} = V_{C2} + V_L = \frac{1}{1-D_1}V_L = \frac{1}{n}V_H \tag{11}$$

$$V_{S4,stress} = V_H = \frac{n}{(1-D_1)}V_L \tag{12}$$

On the basis of the equivalent circuit at time $(1-D_1)T_S$, the voltage stress of $S_1$ is $V_{C1}$, and the voltage across $S_5$ is $V_H$. The voltage stresses of the switches at the aforementioned time are expressed as follows:

$$V_{S1,stress} = V_{C1} = \frac{1}{1-D_1}V_L = \frac{1}{n}V_H \tag{13}$$

$$V_{S5,stress} = V_H = \frac{n}{(1-D_1)}V_L \tag{14}$$

*3.2. Step-Down Mode*

(1)    *Voltage Gain Analysis*

On the basis of KVL, the voltage of the magnetizing inductance $V_{Lm1}$ for time $D_3T_S$ can be expressed as follows:

$$V_{Lm1} = \frac{V_H - V_{c4}}{n} = V_{c3} - V_{c2} + \frac{V_H - V_{c4}}{n} - V_{c1} + V_L = L_{m1}\frac{\Delta i_{Lm1,on}}{D_3T_S} \tag{15}$$

For time $(1-D_3)T_S$, the voltage of the magnetizing inductance $V_{Lm1}$ can be expressed as follows:

$$V_{Lm1} = V_L - V_{c1} = -V_{c2} = \frac{V_{c4}}{n} = \frac{V_{c4}}{n} - V_{c3} = L_{m1}\frac{\Delta i_{Lm1,off}}{(1-D_3)T_S} \tag{16}$$

When volt–second balance is achieved for the inductor at time $(1-D_3)T_S$, the following equation is satisfied:

$$\Delta i_{Lm1,on} = \Delta i_{Lm1,off} \tag{17}$$

By substituting (15) and (16) into (17), the voltages of $C_1$, $C_2$, $C_3$, and $C_4$ can be determined using the following equations:

$$V_{C1} = \frac{1}{(1-D_3)}V_L \tag{18}$$

$$V_{C2} = \frac{D_3}{(1-D_3)}V_L \tag{19}$$

$$V_{C3} = \frac{2D_3}{(1-D_3)}V_L \tag{20}$$

$$V_{C4} = V_H D_3 \tag{21}$$

By substituting (21) into (15), the voltage gain in the step-up mode ($G_{step-down}$) is obtained as follows:

$$G_{step-down} = \frac{V_L}{V_H} = \frac{(1-D_3)}{n} \tag{22}$$

Figure 8 illustrates the relationship between the voltage gain and the duty cycle in the step-down mode.

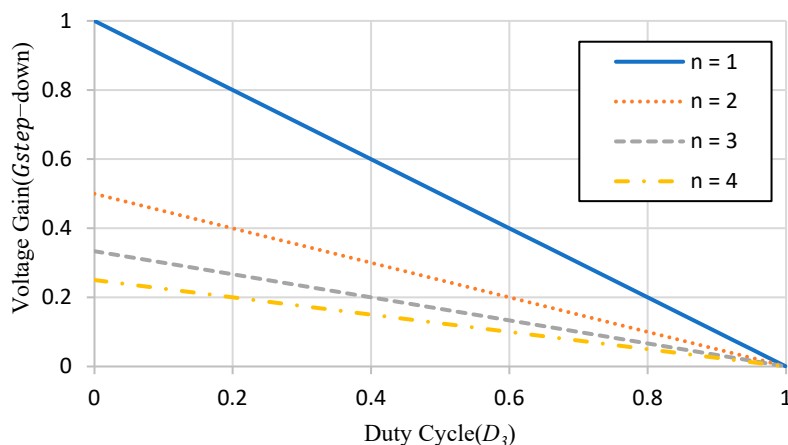

**Figure 8.** Plot of the voltage gain versus the duty cycle in the step-down mode.

(2)  *Voltage Stress Analysis*

According to the equivalent circuit at time $D_3 T_S$, the voltage across $S_2$ is $V_{C1}$, and the voltage across $S_3$ is the sum of $V_{C2}$ and $V_L$. The voltage stress of $S_4$ is $V_H$. The voltage stresses of the switches at the aforementioned time are expressed as follows:

$$V_{S2,stress} = V_{C1} = \frac{1}{n} V_H = \frac{1}{1 - D_3} V_L \tag{23}$$

$$V_{S3,stress} = V_{C2} + V_L = \frac{1}{n} V_H = \frac{1}{1 - D_3} V_L \tag{24}$$

$$V_{S4,stress} = V_H = \frac{n}{(1 - D_3)} V_L \tag{25}$$

According to the equivalent *circuit* at time $(1 - D_3)T_S$, the voltage across $S_1$ is $V_{C1}$, and the voltage stress of $S_5$ is $V_H$. The voltage stresses of the switches at the aforementioned time are expressed as follows:

$$V_{S1,stress} = V_{C1} = \frac{1}{n} V_H = \frac{1}{1 - D_3} V_L \tag{26}$$

$$V_{S5,stress} = V_H = \frac{n}{(1 - D_3)} V_L \tag{27}$$

*3.3. Magnetic Component Design*

(1)  *Step-Up Mode*

In the step-up mode, the magnetic components of the proposed converter are designed to operate in the CCM, and the maximum current of $L_{m1}$ is determined using the following equation:

$$i_{Lm1,max} = i_{Lm1,avg} + \frac{\Delta i_{Lm1}}{2} \tag{28}$$

The minimum current of $L_{m1}$ is expressed as follows:

$$i_{Lm1,max} = i_{Lm1,avg} - \frac{\Delta i_{Lm1}}{2} \tag{29}$$

When the magnetic components are operated in the boundary conduction mode (BCM), the currents $i_{Lm1, min}$ are 0. The current $i_{Lm1, min}$ is expressed as follows:

$$i_{Lm1,min} = 0 = \frac{n}{1 - D_1} I_H - \frac{(1 - D_1)D_1}{2L_{m1}f_s N} V_H \tag{30}$$

After (30) is simplified, $L_{m1,BCM}$ is obtained as follows:

$$L_{m1,BCM} = \frac{(1 - D_1)^2 D_1}{2f_s n^2} \frac{V_H}{I_{H,BCM}} \tag{31}$$

The circuit parameters in the step-up mode are as follows: $V_H$ = 400 V, turn ratio $n$ = 4, switching frequency = 40 kHz, and high-voltage-side current $I_H$ = 0.375 A.

By substituting the aforementioned parameters into (31), the curve of $L_{m1}$ operated in BCM can be plotted (Figure 9). When the inductance of $L_{m1}$ is higher than that in the $L_{m1}$ curve for the BCM, $L_{m1}$ operates in the CCM.

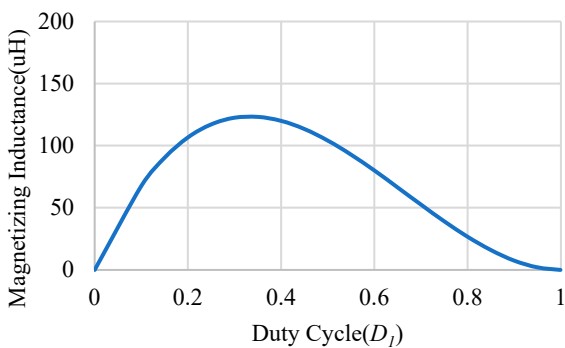

**Figure 9.** $L_{m1, BCM}$ in the step-up mode.

(2)　*Step-Down Mode*

In the step-down mode, the magnetic components of the proposed converter are designed to operate in the CCM, and the maximum current of $L_{m1}$ is determined as Equations (28) and (29).

When the magnetic components are operated in the BCM, the current $i_{Lm1, min}$ is 0. This current can be expressed as follows:

$$i_{Lm1,min} = 0 = I_L - \frac{D_3}{2L_{m1}f_s} V_L \tag{32}$$

After (32) is simplified, $L_{m1,BCM}$ can be obtained as follows:

$$L_{m1,BCM} = \frac{D_3}{2f_s} \frac{V_L}{I_{L,BCM}} \tag{33}$$

The circuit parameters in the step-down mode are as follows: $V_L$ = 48 V, $n$ = 4, switching frequency = 40 kHz, and low-voltage-side current $I_L$ = 3.125 A.

By substituting these parameters into (33), the curve of $L_{m1}$ operated in the BCM can be plotted (Figure 10). When the inductance of $L_{m1}$ is greater than that in the $L_{m1}$ curve for the BCM, $L_{m1}$ operates in the CCM.

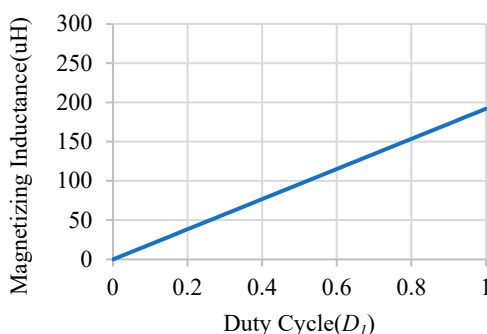

**Figure 10.** $Lm_{1, BCM}$ in step-down mode.

## 4. Experimental Results

The measured waveforms were used to verify the feasibility of the proposed topology. The key waveforms in the step-up and step-down modes were measured separately. Finally, the conversion efficiency of the proposed topology in the step-up and step-down modes were measured. Table 1 shows the electrical specifications and component parameters of the proposed topology, and a photograph of the proposed bidirectional isolated DC–DC converter is displayed in Figure 11.

**Table 1.** Electrical Specifications of the Proposed Topology.

| Parameter | Specification |
|---|---|
| High-side power $P_H$ | 500 W |
| Low-side power $P_L$ | 500 W |
| High-side voltage $V_H$ | 400 V |
| High-side current $I_H$ | 1.25 A |
| Low-side voltage $V_L$ | 48 V |
| Low-side current $I_L$ | 10.416 A |
| Switching frequency $f_s$ | 40 kHz |
| Power switches $S_1$, $S_2$ and $S_3$ | IRFP4568 |
| Power switches $S_4$ and $S_5$ | IXFH60N50P3 |
| Magnetizing inductance $L_{m1}$ | 200 μH |
| Leakage inductance $L_{lk1}$ and $L_{lk2}$ | 2 μH |
| Capacitor $C_1$, $C_2$, $C_3$ and $C_4$ | 50 μF |
| Turns ratio $n$ | 4 |

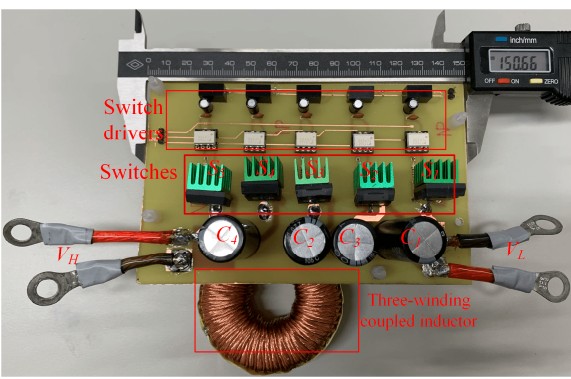

**Figure 11.** Photograph of the proposed converter.

Figure 12a–e shows the key waveforms measured in the step-up mode. Figure 12a displays the measured waveforms of the complementary signals $v_{gs1}$ and $v_{gs2,3}$ and the leakage inductance currents $i_{Llk1}$ and $i_{Llk2}$. Figure 12b depicts the measured waveforms of the voltage and current of the switches $S_1$ and $S_2$. These switches exhibited ZVS in the step-up mode, and their voltage stress was 100 V. Figure 12c illustrates the measured waveforms of the voltage and current of $S_3$ and of the output voltage $V_H$. The switch $S_3$ exhibited ZVS in the step-up mode, and its voltage stress was 100 V. Figure 12d shows the measured waveforms of the voltage and current of $S_4$ and $S_5$. The voltage stress of these switches was 400 V. Figure 12e depicts the measured waveforms of the capacitors $C_1$–$C_4$.

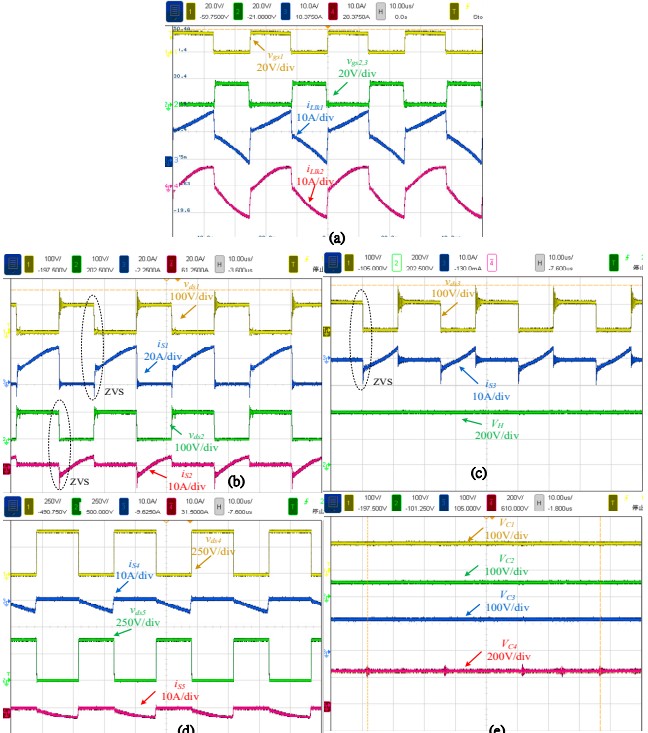

**Figure 12.** Experimental results obtained when operating the proposed converter at full load and a $V_L$ value of 48 V in the step-up mode: (**a**) waveforms of $v_{gs1}$, $v_{gs2,3}$, $i_{Llk1}$, and $i_{Llk2}$; (**b**) waveforms of the $v_{ds}$ and $i_s$ values of $S_1$ and $S_2$; (**c**) waveforms of the $v_{ds}$ and $i_s$ values of $S_3$ and of the output voltage; (**d**) waveforms of the $v_{ds}$ and $i_s$ values of $S_4$ and $S_5$; and (**e**) waveforms of the voltages of $C_1$, $C_2$, $C_3$, and $C_4$.

Figure 13a–e displays the key waveforms measured in the step-down mode. Figure 13a shows the measured waveforms of the complementary signals $v_{gs1,5}$ and $v_{gs2,3,4}$ and the leakage inductance currents $i_{Llk1}$ and $i_{Llk2}$. Figure 13b depicts the measured waveforms of the voltage and current of $S_1$ and $S_2$. The voltage stress of $S_1$ and $S_2$ was 100 V. Figure 13c displays the measured waveforms of the voltage and current of $S_3$ and of the output voltage $V_H$. The voltage stress of $S_3$ was 100 V, and the output voltage of the converter was 48 V. Figure 13d shows the measured waveforms of the voltage and current of $S_4$ and $S_5$. These switches exhibited ZVS, and their voltage stress was 400 V. Figure 13e displays the measured waveforms of $C_1$–$C_4$.

Figure 14 displays the conversion efficiency of the proposed converter in the step-up and step-down modes. The highest conversion efficiency in the step-up mode was 97.59% at 150 W, and the conversion efficiency at full load was 95.03%. The highest conversion efficiency in the step-down mode was 96.5% at 100 W, and the conversion efficiency at full load was 94.08%.

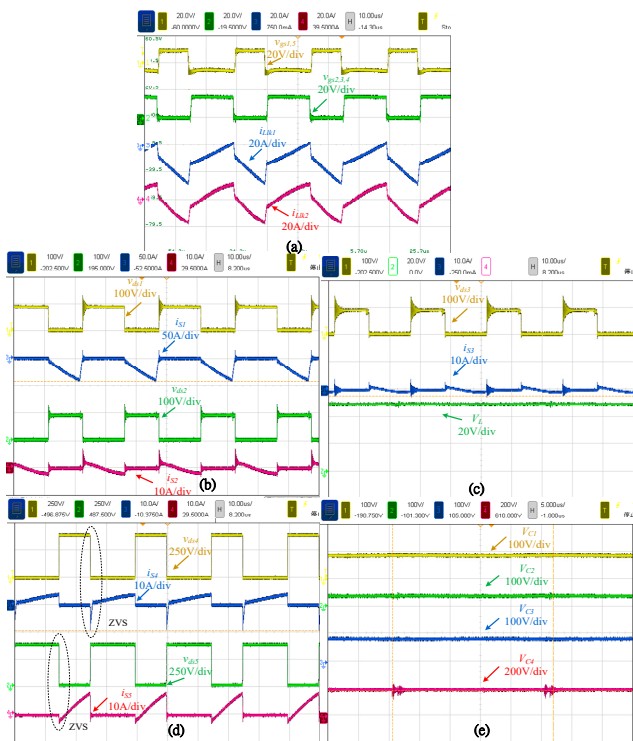

**Figure 13.** Experimental results obtained when operating the proposed converter at full load and a $V_H$ value of 400 V in the step-down mode: (**a**) waveforms of $v_{gs1,5}$, $v_{gs2,3,4}$, $i_{Llk1}$, and $i_{Llk2}$; (**b**) waveforms of the $v_{ds}$ and $i_s$ values of $S_1$ and $S_2$; (**c**) waveforms of the $v_{ds}$ and $i_s$ values of $S_3$ and of the output voltage; (**d**) waveforms of the $v_{ds}$ and $i_s$ values of $S_4$ and $S_5$; and (**e**) waveforms of the voltages of $C_1$, $C_2$, $C_3$, and $C_4$.

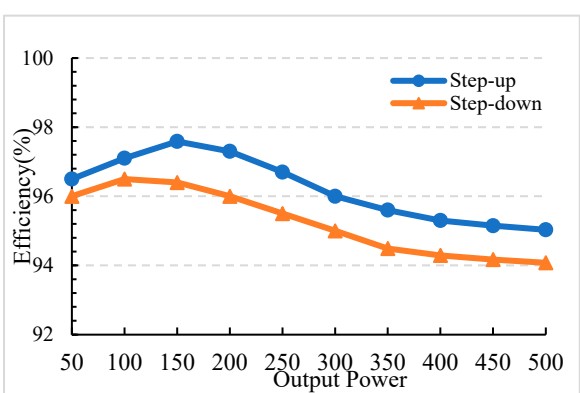

**Figure 14.** Efficiency of the proposed converter.

Table 2 presents a comparison between the proposed converter and those developed in [20,21,24,27,30]. In general, the proposed converter has fewer components, a higher efficiency, and a higher voltage gain than do the other converters, which indicates that the proposed converter can be widely used in numerous industries. Among the compared converters, the converter developed in [24] has the fewest components but has a low voltage gain and the lowest output power. Figures 15 and 16 depict the voltage gain of the compared converters in the step-up and step-down modes, respectively. The converter developed in [27] has the highest voltage gain among all the compared converters but has a lower efficiency and a higher number of components than does the converter proposed in this paper. The converter in [30] has fewer switches but lower efficiency.

**Table 2.** Comparison between the proposed converter and related bidirectional converters.

| | Converter in [20] | Converter in [21] | Converter in [24] | Converter in [27] | Converter in [30] | Proposed Converter |
|---|---|---|---|---|---|---|
| $G_{step-up}\left(\frac{V_H}{V_L}\right)$ | $\frac{2+D}{(1-D)}$ | $\frac{3}{(1-D)}$ | $\frac{3D+n}{n(1-D)}$ | $\frac{n}{(1-D)^2}$ | $\frac{n}{(1-D)}$ | $\frac{n}{(1-D)}$ |
| $G_{step-down}\left(\frac{V_L}{V_H}\right)$ | $\frac{D}{(3-D)}$ | $\frac{D}{3}$ | $\frac{nD}{n+3(1-D)}$ | $\frac{(1-D)^2}{n}$ | $\frac{1-D}{n}$ | $\frac{1-D}{n}$ |
| $V_L$ | 40–120 V | 30–100 V | 48 V | 24–48 V | 48 V | 48 V |
| $V_H$ | 400 V | 400 V | 120 V | 400 V | 400 V | 400 V |
| Switches | 5 | 8 | 4 | 6 | 4 | 5 |
| Magnetic Components | 2 | 3 | 2 | 2 | 1 | 1 |
| Capacitors | 6 | 6 | 0 | 3 | 4 | 4 |
| Diodes | 0 | 0 | 1 | 0 | 0 | 0 |
| Efficiency of step-up mode | 94.09% | 95.8% | 98% | 95.6% | 96.8% | 97.59% |
| Efficiency of step-down mode | 94.41% | 95.9% | 97% | 94.2% | 95.2% | 96.5% |
| Isolated | No | No | No | Yes | Yes | Yes |

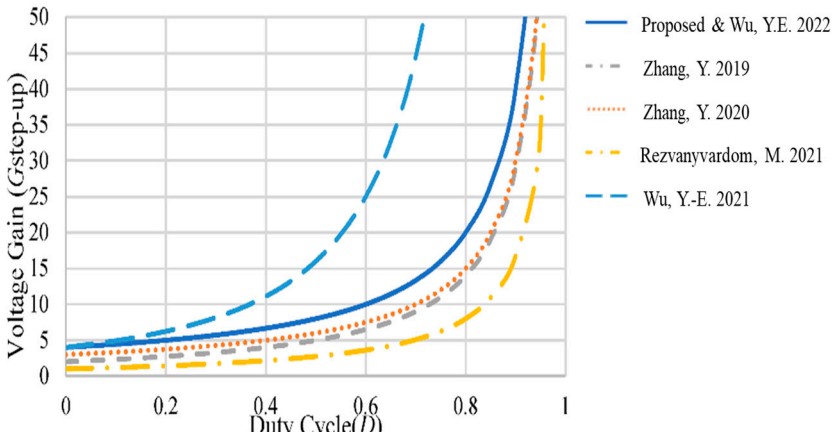

**Figure 15.** Voltage gain of the compared converters in the step-up mode [20,21,24,27,30].

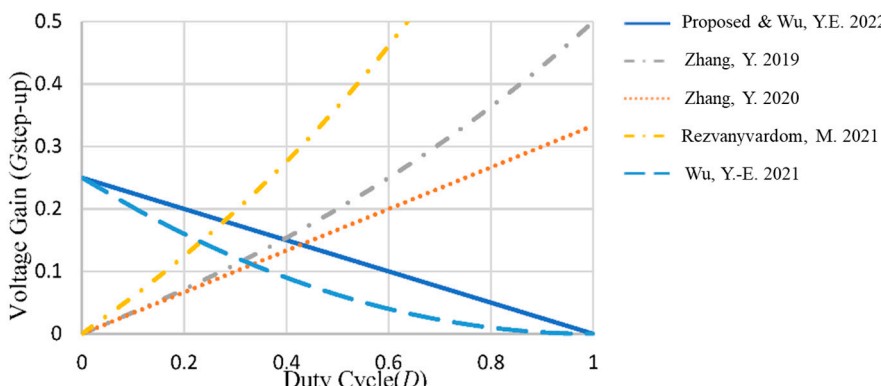

**Figure 16.** Voltage gain of the compared converters in the step-down mode [20,21,24,27,30].

Figures 17 and 18 depict the conversion efficiency of the compared converters in the step-up and step-down modes, respectively. The converter developed in [20] does not have a high conversion efficiency and has insufficient voltage gain. The converter developed in [21] has the highest output power but requires a highly complex control method. The converter developed in [24] has the highest efficiency but has a low output power and insufficient voltage gain. The converter developed in [27] has the highest voltage gain but has insufficient efficiency when the load level is less than half the full load and contains a

high number of components. The converter in [30] has fewer components, but efficiency is lower. Finally, the converter proposed in this paper has a high conversion efficiency and voltage gain in the step-up and step-down modes.

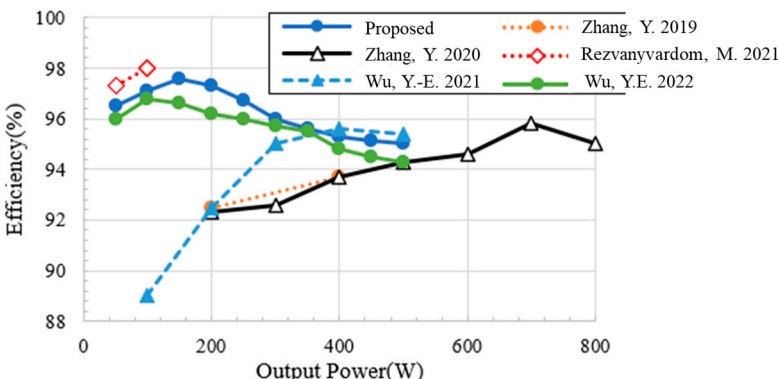

**Figure 17.** Efficiency of the compared converters in the step-up mode [20,21,24,27,30].

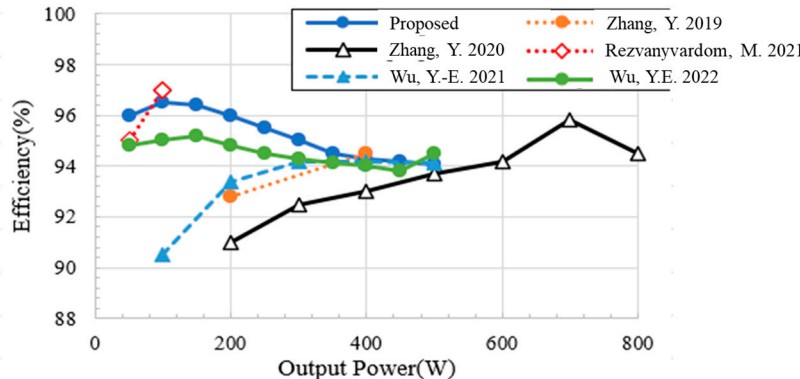

**Figure 18.** Efficiency of the compared converters in the step-down mode [20,21,24,27,30].

## 5. Conclusions

In this paper, a novel high-efficiency bidirectional isolated DC–DC converter is proposed for an energy storage system. This converter only requires one complementary PWM signal to control the step-up and step-down modes. Partial switches are used in this converter to achieve ZVS, which can increase the converter efficiency. The feasibility of the proposed topology was examined through theoretical analysis, simulations, and experiments. In the experiments, the maximum efficiency of the proposed converter in the step-up and step-down modes was 97.59% and 96.5%, respectively.

In conclusion, the advantages of the proposed converter are as follows: (1) it has a simple circuit structure, (2) it only requires one complementary PWM signal to control the step-up and step-down modes, (3) it achieves a high voltage gain and high efficiency, (4) its input and output power supplies are separated through galvanic isolation, (5) the current of its low-voltage side is continuous in the CCM, and (6) ZVS is achieved using specific switches to reduce its switching loss.

**Author Contributions:** Conceptualization, Y.-E.W.; methodology, Y.-E.W. and K.-C.C.; formal analysis, Y.-E.W. and K.-C.C.; investigation, K.-C.C.; resources, Y.-E.W.; writing—original draft preparation, Y.-E.W. and K.-C.C.; writing—review and editing, Y.-E.W.; project administration, Y.-E.W.; funding acquisition, Y.-E.W. All authors have read and agreed to the published version of the manuscript.

**Funding:** This research received no external funding.

**Institutional Review Board Statement:** Not applicable.

**Informed Consent Statement:** Not applicable.

**Data Availability Statement:** All data utilized in this study are available online.

**Conflicts of Interest:** The authors declare no conflict of interest.

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
