# Peer review of "High Efficiency and High Voltage Conversion Ratio Bidirectional Isolated DC–DC Converter for Energy Storage Systems"

_processes, doi:10.3390/pr10122711_

Round 1
Reviewer 1 Report
This presents a bi-directional isolated DC–DC converter. It achieves a higher efficiency with fewer components with a wide input-output voltage range. The paper is clearly written and easy to follow. The theoretical and experimental results are clear. Several suggestions for improvement:
1. The design depends heavily on PWM waveform synchronization and timing control. However, looking at the layout design, the switches are linearly placed, resulting in a wide difference in terms of latency mismatch in the control signals. The paper needs to further consider the parasitics of the board design and clearly study the impact of the unbalanced/skewed control signals.
2. Compared to [27], the improvement is not very significant. This paper needs a careful explanation of the difference and tradeoffs between these two works.
3. How are figures 3 and 5 obtained? It seems a basic circuit without any major parasitics is considered.
4. I would suggest adding a power density factor in Table 2 for comparison. In general, transistor number is not critical, but reducing passive sizes is more critical for high-density applications.
5. Overall, the reviewer feels this is incremental work but has good value for the community. Adding more thorough discussions and comparisons in the introduction is critical. Currently, the writing only scratches the surface on each previous work.
Author Response
This presents a bi-directional isolated DC–DC converter. It achieves a higher efficiency with fewer components with a wide input-output voltage range. The paper is clearly written and easy to follow. The theoretical and experimental results are clear. Several suggestions for improvement:
- The design depends heavily on PWM waveform synchronization and timing control. However, looking at the layout design, the switches are linearly placed, resulting in a wide difference in terms of latency mismatch in the control signals. The paper needs to further consider the parasitics of the board design and clearly study the impact of the unbalanced/skewed control signals.
Author response:
Thank you for reviewer’s comments, the control in this paper is currently using a signal generator to generate a PWM signal for open-loop control, and there is no requirement for the layout loop of the control signal. However, the points mentioned by the reviewer are areas that can continue to be improved in the future.
- Compared to [27], the improvement is not very significant. This paper needs a careful explanation of the difference and tradeoffs between these two works.
Author response:
Thank you for reviewer’s comments. Compared with [27], this paper uses fewer switching elements and magnetic elements, and achieves higher efficiency.
- How are figures 3 and 5 obtained? It seems a basic circuit without any major parasitics is considered.
Author response:
Thank you for reviewer’s comments. Figure 3 and Figure 5 are the key current and voltage waveform diagrams of the circuit, which are drawn through the simulation software and the implemented measured results of the circuit.
- I would suggest adding a power density factor in Table 2 for comparison. In general, transistor number is not critical, but reducing passive sizes is more critical for high-density applications.
Author response: Thank you for reviewer’s comments. I believe it is good to add power density for comparison. However, I cannot know the power density of other paper. I can only calculate its power density in general through the physical photographs in these paper, which will lose the fairness of the comparison.
- Overall, the reviewer feels this is incremental work but has good value for the community. Adding more thorough discussions and comparisons in the introduction is critical. Currently, the writing only scratches the surface on each previous work.
Author response:
Thank you for reviewer’s comments, the revised paper [30] supplements another paper, and compares and discusses its advantages and disadvantages.
Response:
Thanks for reviewer's comments and guidance. The authors try to reply to the problems of reviewer, please reviewer tolerant if there are any incomplete.
Reviewer 2 Report
Authors discussed their work on the development of the Bidirectional Isolated DC–DC Converter having High Efficiency and High Voltage Conversion ratio which can be used for Energy Storage SystemsThe following are the major queries related to present work and I must reject this paper based on them.
1. There are similarities between the present work and existing published work in the literature.
Following is the already published work journal publication.
Y. E. Wu and B. H. Pan, "High Efficiency and Voltage Conversion Ratio Bidirectional Isolated DC-DC Converter for Energy Storage System,"
in IEEE Access, vol. 10, pp. 55187-55199, 2022, doi: 10.1109/ACCESS.2022.3177206.
Following are the specific similarities:
1. Fig.1-5 (published work) and Fig.1-5 (present work) are the same.
2.The outcome of the present and published work is similar.
Authors must clarify/justify if the present work is similar to already published work or not.
2. Authors have claimed the reduction of the switching losses are less in their Bidirectional Isolated DC–DC Converter.
However, Iron and ohmic losses are not illustrated for the Bidirectional Isolated DC–DC Converter.
3. The error bars in the Fig.17,18 should be added in order to show the variation and standard errors in the output characteristics.
4. First of all, there are few formatting errors and few grammatical mistakes with references. Kindly correct it.
5. Try to add text details in all the figures.
Author Response
Authors discussed their work on the development of the Bidirectional Isolated DC–DC Converter having High Efficiency and High Voltage Conversion ratio which can be used for Energy Storage Systems
The following are the major queries related to present work and I must reject this paper based on them.
1. There are similarities between the present work and existing published work in the literature. Following is the already published work journal publication.
Y. E. Wu and B. H. Pan, "High Efficiency and Voltage Conversion Ratio Bidirectional Isolated DC-DC Converter for Energy Storage System,"
in IEEE Access, vol. 10, pp. 55187-55199, 2022, doi: 10.1109/ACCESS.2022.3177206.
Following are the specific similarities:
1). Fig.1-5 (published work) and Fig.1-5 (present work) are the same.
Response:
Thanks for reviewer's comments. Figure 1 is the most common distributed system architecture diagram, which is used to illustrate the application of bidirectional DC/DC converters, so it is indeed very similar to the previous paper, but the circuit architecture and operation mode diagram in Figure 2-5 are very different from those of the previous paper.
2). The outcome of the present and published work is similar.
Authors must clarify/justify if the present work is similar to already published work or not.
Response:
Thanks for reviewer's comments. We have added a comparison with the previous paper [30] in the revised paper. From the comparison, we can see that although [30] uses fewer parts, the overall efficiency is lower than this paper.
2. Authors have claimed the reduction of the switching losses are less in their Bidirectional Isolated DC–DC Converter. However, Iron and ohmic losses are not illustrated for the Bidirectional Isolated DC–DC Converter.
Response:
Thanks for reviewer's comments. Generally speaking, in the switching circuit architecture, the switching loss is much greater than the ohmic loss such as iron loss. In addition, the current specification in this paper is not too large, and the ohmic loss becomes smaller in the overall loss, so it is not considered in this paper. This paper focus to reduce the switching loss to make an effective contribution for improvement of efficiency.
3. The error bars in the Fig.17, 18 should be added in order to show the variation and standard errors in the output characteristics.
Response:
Thanks for reviewer's comments. Figures 17 and 18 are efficiency comparisons with other papers. Because the electrical specifications of each paper are different, and there are a total of six comparison curves, adding error bars to the graph may make the entire graph more complicated and less readable.
4. First of all, there are few formatting errors and few grammatical mistakes with references. Kindly correct it.
Response:
Thanks for reviewer's comments. It has been corrected in the revised paper.
- Try to add text details in all the figures.
Response:
Thanks for reviewer's comments. It has been improved in the revised paper.
Response:
Thanks for reviewer's comments and guidance. The authors try to reply to the problems of reviewer, please reviewer tolerant if there are any incomplete.
Reviewer 3 Report
A novel high-efficiency bidirectional isolated DC–DC converter that can be applied to an energy storage system for battery charging and discharging is proposed. The proposed topology comprises five switches and a common core coupled inductor. Switches exhibit ZVS, which reduces the switching. The experimental results indicate that the highest efficiency of the proposed converter in the step-up and step-down modes is 97.59% and 96.5%, respectively.
My comments are as follows:
Increasing the efficiency of power supplies is of significant importance. The author shows multiple results — the experimental waveforms at different powers.
1- In Figures 17 and 18, How the efficiency was measured and calculated?
2- Discussion on the limitation of the proposed technique should be added.
3- What are the exactions of the author when the power is increased by more than 500 W, e.g. in terms of efficiency?
Author Response
A novel high-efficiency bidirectional isolated DC–DC converter that can be applied to an energy storage system for battery charging and discharging is proposed. The proposed topology comprises five switches and a common core coupled inductor. Switches exhibit ZVS, which reduces the switching. The experimental results indicate that the highest efficiency of the proposed converter in the step-up and step-down modes is 97.59% and 96.5%, respectively.
My comments are as follows:
Increasing the efficiency of power supplies is of significant importance. The author shows multiple results — the experimental waveforms at different powers.
1- In Figures 17 and 18, How the efficiency was measured and calculated?
Authors Response
Thanks for reviewer’s comments. The input power is obtained through the built-in wattmeter of the power supply, the output power is obtained through the built-in wattmeter of the electronic load, and the efficiency is obtained by dividing the two.
2- Discussion on the limitation of the proposed technique should be added.
Authors Response
Thanks for reviewer’s comments. The structure of this paper uses a three-winding coupled inductor. To achieve higher wattage and higher voltage gain, thicker copper wires and higher turn ratios are required, which will greatly increase the size of the coupled inductor and the overall circuit volume.
3- What are the exactions of the author when the power is increased by more than 500 W, e.g. in terms of efficiency?
Authors Response
Thanks for reviewer’s comments. The coupled inductor in this paper is designed for a full load of 500W. When it exceeds 500W, a larger duty cycle is required to stabilize the voltage, and the measured efficiency will drop significantly when it is over 600W.
Response:
Thanks for reviewer's comments and guidance. The authors try to reply to the problems of reviewer, please reviewer tolerant if there are any incomplete.
Round 2
Reviewer 2 Report
The authors have taken all the comments into the account sincerely and modified the manuscript which improved the overall quality. Authors have cited as well as compared their previously published work. Hence, the manuscript can be accepted in its present form.